# Association of G-Protein-Coupled Receptors autoantibodies with vasoregulation in Post-COVID

Felix S. Seibert[ID][1]*, Melisa Kurucay[ID][1], Lea Wiemers[ID][1], Ulrik Stervbo[ID][1], Oliver Sander[ID][2], Monika Segelbacher[3], Maximilian Seidel[1], Sebastian Bertram[1], Nina Babel[1], Timm H. Westhoff[1]

1 Medical Department 1, University Hospital Marien Hospital Herne, Ruhr-University Bochum, Bochum, Germany, 2 Department and Hiller Research Unit of Rheumatology, Medical Faculty, University Hospital Düsseldorf, Heinrich-Heine-University Düsseldorf, Düsseldorf, Germany, 3 Department of Internal Medicine, Gastroenterology and Pneumology, Marien-Hospital Witten, Witten, Germany

* felix.seibert@elisabethgruppe.de

## Abstract

### Background

The Post-COVID syndrome is associated with the generation of autoantibodies to vasoregulative G-protein coupled receptors (GPCR). It remains elusive, however, whether these autoantibodies play a pathophysiological role in this disease. The present study investigates whether detection and concentration of GPCR autoantibodies are related to vascular function in patients with Post-COVID.

### Materials and methods

We performed a cross-sectional study, enrolling 80 patients with Post-COVID and 54 individuals with a history of SARS-CoV-2 infection without persisting symptoms (control group). ELISA measurement of GPCR autoantibodies encompassed auto-antibodies to Angiotensin-II-Receptor-1 (AGTR2), Beta-1 Adrenergic Receptor (ADRB1), Beta-2 Adrenergic Receptor (ADRB2), Endothelin Receptor (EDNRA), Muscarinergic Choline Receptor 3 (CHRM3), and Muscarinergic Choline Receptor 4 (CHRM4). Endothelium-dependent vasodilation was assessed by flow mediated dilation (FMD). Measurement of central aortic blood pressure and capillary nailfold capillary microscopy were performed as additional assessments of vasoregulatory function. Lipoprotein-associated phospholipase A2 (Lp-PLA2) served as markers of vascular inflammation.

### Results

52 (65%) patients with Post-COVID had positive autoantibody findings above previously established cut-off values, the incidence was lower in the non-Post-COVID

**Data availability statement:** The dataset is publicly available and can be accessed at the following DOI: https://doi.org/10.6084/m9.figshare.30127723.

**Funding:** The author(s) received no specific funding for this work.

**Competing interests:** The authors have declared that no competing interests exist.

group (n = 12, 22.2%, p = 0.0001). The median concentrations for AGTR2, ADRB1, CHRM3 and CHRM4 autoantibodies were significantly higher in the symptomatic cohort (p < 0.05 each). Spearman correlation analysis showed a strong and significant negative correlation of several GPCR autoantibodies with aortic systolic blood pressure (AGTR2 p = 0.026, ADRB1 p = 0.001, ADRB2 p = 0.012) and aortic diastolic blood pressure (ADRB1 p = 0.005, CHRM4 p = 0.046) in Post-COVID. High EDNRA autoantibodies titers were associated with FMD (p = 0.038). There was no significant association of any GPCR autoantibody concentration with FMD or Lp-PLA2 in the control group.

## Conclusion

GPCR autoantibodies were highly prevalent in this Post-COVID cohort. Several GPCR autoantibodies were associated with measures of vasorelaxation like lower systolic and diastolic aortic blood pressure and stronger endothelium-dependent vasodilation. However, given the absence of differences in microvascular and macrovascular function, the precise role of GPCR autoantibodies remains elusive.

## Introduction

In contrast to a decreasing number of severe acute courses of COVID-19, the incidence of Post-COVID remains at a high level, going along with a significant socio-economic impact. About 5% of all SARS-CoV-2-infected patients do not completely recover from COVID-19 and fulfill the criteria of Post-COVID [1–3]. The Post-COVID syndrome is defined by the WHO as the continuation or the development of not otherwise explainable new symptoms after the onset of an acute SARS-CoV-2-infection for a duration of at least two months. Post-COVID presents with a variety of multiorgan disorders, including fatigue, post exertional malaise, shortness of breath, myalgia and cognitive impairment. Post-COVID shows striking similarities with the encephalomyelitis/chronic fatigue syndrome (ME/CFS) and is increasingly recognized as a SARS-CoV-2 related ME/CFS variant [4].

The pathophysiology of both of the above-mentioned diseases is incompletely understood and currently under investigation. Postinfectious persisting tissue damage, autoimmunity, vascular dysfunction and virus persistence are four prominent hypotheses in this context [5]. We recently linked the generation of autoantibodies against vasoregulatory G-protein coupled receptors (GPCR) to the intensity of neurological disorders including psychomotor speed, visual search, attention, and fatigue [6].

The hypothesis of impaired micro- and macrovascular dysregulation following SARS-CoV-2 infection, results from a sound body of evidence [7]. The underlying mechanisms, however, remain elusive. The present study investigates whether the generation of autoantibodies to GPCR is associated with alterations in vasoregulatory function.

## Materials and methods

### Protocol and patients

We performed a monocentric cross-sectional study at the University Hospital of the Ruhr-University Bochum, Germany, enrolling 80 patients with Post-COVID. Post-COVID was diagnosed according to the German S1 guideline [8]. All neuro-psychological assessments were performed applying to the German version of the "Consortium to Establish a Registry for Alzheimer's Disease Neuropsychological Assessment Battery" [9]. Enrollment encompassed mild to moderate SARS-CoV-2 infection up to 12 months prior to inclusion. A second group (n = 54) of individuals had SARS-CoV-2 infection, but without presenting ongoing symptoms (control group). All patients signed informed consent. The Ethics Committee of the Ruhr-University of Bochum approved the study (ID 22–7491) in accordance with the 1964 Declaration of Helsinki and its later amendments.

### Measurement of autoantibodies to GPCR

The GPCR autoantibody panel was selected based on prior biological relevance and established use in the literature. We applied a commercially available GPCR autoantibody panel that has been used in our previous work [6]. GPCR autoanti-bodies encompassed autoantibodies against Angiotensin-II-Receptor-1 (AGTR2), Beta-1 Adrenergic Receptor (ADRB1), Beta-2 Adrenergic Receptor (ADRB2), Endothelin Receptor (EDNRA), Muscarinergic Choline Receptor 3 (CHRM3), and Muscarinergic Choline Receptor 4 (CHRM4), all measured by ELISA. Serum was collected in a S-Monovette Serum collection tube (Sarstedt, Germany) and processed per manufacturer's instructions. Purified serum was stored at −20°C until use. ELISA systems for detection of autoantibodies against AGTR2, ADRB1, ADRB2, EDNRA, CHRM3, and CHRM4 were all obtained from CellTrend, Germany. Autoantibodies in serum were determined per manufacturer's instructions and normalized using the provided standard. Cut-offs for the detection have been chosen according to the manufacturer. The following cut-offs were used to define presence of autoantibodies: AGTR2: > 10 U/ml; ADRB1: > 15 U/ml; ADRB2: > 8 U/ml; EDNRA: > 10 U/ml; CHRM3: > 10 U/ml; CHRM4: > 6 U/ml.

### Measurement of markers of vascular and systemic inflammation

Vascular inflammation was assessed by measurement of lipoprotein-associated phospholipase A2 (Lp-PLA2). Lp-PLA2 is regarded as a marker of vascular inflammation [10]. Moreover, Lp-PLA2 has been found to be correlated with several comorbidities, complications and the severity of COVID-19 [11]. Both parameters were assessed according to the manu-facturer's protocol (kit number L26722 [Asbach Medical] and 09877 [Roche], respectively)

### Measurement of endothelium-dependent vasodilation by flow-mediated dilation

Endothelium-dependent vasodilation was assessed by flow mediated dilation (FMD). Vascular endothelial function of the right brachial artery was studied with ultrasonography (Aloka/Hitachi, Prosound alpha 6) with a high-frequency (5–13 MHz) linear-array transducer. The examinations were performed under standard conditions (room temperature 22 °C, quiet environment). Blood pressure was measured by an oscillometric method before initiation of FMD measurement. Par-ticipants were positioned in a supine position for 15 min to avoid a potential effect of stress and they were instructed not to speak during the examination. Assessment of endothelium-dependent vasodilation was done at the level of right brachial artery 5–10 cm above the antecubital fossa, according to the recommendations for FMD assessment [12,13].

The tracking gate followed the motion of the vessel walls caused by pulsations and automatically measured the change in vessel diameter with a precision up to 0.01 mm in real time. The waveform of diameter changes over the cardiac cycle was displayed in real time using the FMD-mode of the eTRACKING system. The internal diameter of right brachial artery was continuously monitored using following protocol: 1 min recording of baseline diameter at rest, 5 min recording during a forearm ischemia induced by inflation of a pneumatic forearm cuff 50 mmHg above systolic blood pressure, and 3 min of

post-deflation diameter recording. All the examinations were performed by one and the same specialist trained in 2D and Doppler ultrasonography. To minimize operator-dependent error, a mechanical probe holder was used with the fixed angle approximately 60° between the probe and the vessel orientation in all the examinations.

### Central aortic blood pressure

We performed a non-invasive pulse wave analysis to investigate central aortic blood pressure. The Mobil-O-Graph device performs an oscillometric assessment of peripheral blood pressure and uses tonometric contour analysis of the brachial arterial pulse wave in order to provide systolic, diastolic and mean blood pressure values after a stepwise deflation process of the cuff. Calibration is performed using systolic and diastolic pressure. This system calculates central blood pressure, aortic PWV and additional central hemodynamic indices, all based on the oscillometric recording of pulse waves at the brachial artery site. In a first step, the recorded pulse waves are checked for their plausibility and qualified according to pre-defined quality criteria. Once the input signal has qualified as acceptable, the concepts of wave harmonics and Fourier analysis are applied for the calculation of an aortic pressure curve. The device has been validated by a comparison to intraarterial measurements [14,15] and an established tonometric method [16,17]. Central blood pressure measurement was not available in the control group.

### Nailfold capillary microscopy

Standardized capillary microscopic examinations of the nailfolds of both hands (digits II-V) were performed on patients diagnosed with Post-COVID. The images (close-up with details representing 1 mm/210-fold magnification for each patient) were pseudonymized and stored. The images were assessed according to standardized criteria with regard to capillary density and morphological changes including vessel diameter variability, as well as elongation and ramification (neoangiogenesis). The lower cut-off for normal density was 7/mm. Nailford capillary microscopy was not performed in the control group.

### Statistics

Data were analyzed for Gaussian distribution (D'Agostino Pearson). In case of normal distribution, data are presented as mean with standard deviation (SD), otherwise, data are given as median with interquartile range (IQR). Differences between groups were assessed using the Mann-Whitney-U-test. If relevant, Spearman correlation was performed.

### Results

Patient demographics of both cohorts are summarized in Table 1. With 45 years, both groups were of similar age. Sex distribution analysis revealed that significantly more women were affected in the Post-COVID cohort (72.5%, p = 0.0022). BMI was lower for participants without Post-COVID (24.9 ± 3.34 vs. 27.6 ± 5.9, p = 0.0214). In our cohort, 40 out of 80 patients with Post-COVID (50%) had received at least one SARS-CoV-2 vaccination prior to study inclusion, whereas 48 out of 54 individuals (88.9%) in the non-Post-COVID group were vaccinated, resulting in the reported significant difference (p = 0.0001; Table 1). Accordingly, 40 patients (50%) in the Post-COVID group and 6 subjects (11.1%) in the control group were unvaccinated. Neurological impairment as assessed via CERAD-Plus was numerically stronger in the Post-COVID group, but without reaching statistical significance (p = 0.1061). Prevalence of cardiovascular comorbidities like diabetes, coronary heart disease, hypercholesterinemia, hypertension and smoking did not significantly differ between the groups. 52 (65%) patients with Post-COVID had positive GPCR autoantibody findings above previously established cut-off values. The median absolute concentrations and the subgroup of patients, presenting at least one positive GPCR autoantibody, are presented in Table 2.

**Table 1. Population characteristics.**

| | | Post-COVID (N = 80) | No Post-COVID (N = 54) | p |
|---|---|---|---|---|
| Age | | 45 ± 13 | 45 ± 19 | 0.9406 |
| Sex | Male | 22 (27.5%) | 30 (55.6%) | **0.0022** |
| | female | 58 (72.5%) | 24 (44.4%) | |
| BMI (kg/m²) | | 27.6 ± 5.9 | 24.9 ± 3.34 | **0.0214** |
| Time from infection to blood sampling in days | | 403 (350-455) | 157 (134-179) | **0.0001** |
| Vaccinated | Yes | 40 (50%) | 48 (88.9%) | **0.0001** |
| | No | 40 (50%) | 6 (11.1%) | |
| Neurological symptoms (CERAD-Plus test battery) | | 134.5 ± 16.8 | 139.6 ± 15.8 | 0.1061 |
| *Comorbidities* | | | | |
| Hypertension | | 29 (36.3%) | 11 (20.4%) | 0.0560 |
| Diabetes | | 7 (8.8%) | 3 (5.6%) | 0.5258 |
| Heart failure | | 3 (3.8%) | 0 (0%) | 0.2693 |
| Coronary heart disease | | 3 (3.8%) | 1 (1.9%) | 0.6443 |
| Chronic obstructive pulmonary disease | | 3 (3.8%) | 0 (0%) | 0.2693 |
| Atrial fibrillation | | 1 (1.3%) | 0 (0%) | 0.9999 |
| Chronic kidney disease | | 0 (0%) | 0 (0%) | 0.9999 |
| Peripheral artery disease | | 0 (0%) | 0 (0%) | 0.9999 |
| Hypercholesterinemia | | 15 (18.8%) | 6 (11.1%) | 0.2362 |
| Smoking | | 12 (15%) | 4 (7.4%) | 0.2774 |

BMI: body mass index. CERAD-Plus: Consortium to Establish a Registry for Alzheimer's Disease Neuropsychological Assessment Battery.

**Table 2. GPCR autoantibody findings and FMD in Post-COVID and no Post-COVID.**

| GPCR auto-ab (U/ml) | Post-COVID | *In case of ≥1 positive GPCR ab* | No Post-COVID | p | *In case of ≥1 positive GPCR ab* | p |
|---|---|---|---|---|---|---|
| AGTR2 | 9.86 (6.47-13.17) | *13.21 (11.57-18.51)* | 7.25 (6.10-9.79) | **0.0071** | *14.52 (11.59-35.68)* | 0.4652 |
| ADRB1 | 9.74 (5.13-15,99) | *19.12 (16.03-27.54)* | 5.53 (2.86-9.57) | **0.0024** | *17.54 (11.82-32.02)* | 0.5973 |
| ADRB2 | 2.83 (1.37-8.39) | *14.18 (10.83-31.26)* | 3.94 (2.91-9.27) | **0.0073** | *40.0 (22.51-110.4)* | **0.0030** |
| EDNRA | 5.74 (4.51-8.24) | *14.07 (11.30-25.19)* | 5.58 (4.61-7.36) | 0.6044 | *24.32 (8.01-24.32)* | 0.4277 |
| CHRM3 | 6.80 (4.01-10.34) | *13.73 (12.04-25.30)* | 4.58 (3.48-6.79) | **0.0210** | *36.81 (7.77-36.81)* | 0.5745 |
| CHRM4 | 5.83 (3.32-9.20) | *9.27 (7.15-13.02)* | 2.16 (1.69-4.04) | **0.0001** | *5.61 (2.95-18.23)* | 0.1475 |
| **FMD (%)** | 5.59 (4.16-8.27) | *5.62 (4.34-8.39)* | 6.77 (4.28-9.00) | 0.2236 | *7.31 (4.64-9.32)* | 0.2775 |

Numbers in brackets indicate interquartile range. ab – antibody, AGTR2 – Angiotensin-II-Receptor-1 ADRB1 – Beta-1 Adrenergic Receptor, ADRB2 – Beta-2 Adrenergic Receptor, EDNRA – Endothelin Receptor, CHRM3 – Muscarinergic Choline Receptor 3, CHRM4 – Muscarinergic Choline Receptor 4. FMD – flow mediated dilation.

Amongst 54 patients without Post-COVID, a significantly lower amount (n = 12, 22.2%, p = 0.001) were positive for at least one GPCR autoantibody. As seen in Table 2, the median concentrations for AGTR2, ADRB1, CHRM3 and CHRM4 autoantibodies were significantly lower in the asymptomatic cohort (p < 0.05 each). No correlation for any GPCR autoantibodies and FMD was seen in patients without Post-COVID (p > 0.05 each, Table 3).

**Table 3. Spearman correlation analysis for hemodynamic parameters and nailfold capillaroscopy.**

| GPCR auto-ab | Post-COVID | | | | | | | No Post-COVID |
|---|---|---|---|---|---|---|---|---|
| | FMD | central systolic BP | Central diastolic BP | Capillary density | Vessel diameter variability | Elongation | Ramification | FMD |
| AGTR2 | ρ=0.086 p=0.4522 | ρ=−0.389 **p=0.0276** | ρ=−0.304 p=0.0912 | ρ=−0.187 p=0.5582 | ρ=−0.317 p=0.3163 | ρ=−0.342 p=0.2745 | ρ=−0.088 p=0.7853 | ρ=0.062 p=0.6571 |
| ADRB1 | ρ=0.053 p=0.6448 | ρ=−0.513 **p=0.0027** | ρ=−0.489 **p=0.0045** | ρ=−0.212 P=0.5057 | ρ=−0.387 p=0.2153 | ρ=−0.455 p=0.1386 | ρ=−0.194 p=0.5425 | ρ=−0.012 p=0.9292 |
| ADRB2 | ρ=0.062 p=0.5873 | ρ=−0.543 **p=0.0013** | ρ=−0.310 p=0.0848 | ρ=−0.187 p=0.5582 | ρ=−0.310 p=0.3274 | ρ=−0.363 p=0.2445 | ρ=−0.035 p=0.9162 | ρ=−0.007 p=0.9629 |
| EDNRA | ρ=0.234 **p=0.0382** | ρ=−0.319 p=0.0753 | ρ=−0.215 p=0.2366 | ρ=0.134 p=0.676 | ρ=−0.387 p=0.2153 | ρ=−0.494 p=0.1048 | ρ=0.170 p=0.5964 | ρ=0.078 p=0.5734 |
| CHRM3 | ρ=0.075 p=0.5125 | ρ=0.009 p=0.9599 | ρ=−0.215 p=0.2378 | ρ=−0.014 p=0.9684 | ρ=−0.063 p=0.8509 | ρ=0.205 p=0.5201 | ρ=0.1873 p=0.5577 | ρ=0.068 p=0.6249 |
| CHRM4 | ρ=0.054 p=0.6373 | ρ=−0.191 p=0.2955 | ρ=−0.456 **p=0.0086** | ρ=−0.194 p=0.5431 | ρ=−0.211 p=0.5105 | ρ=−0.307 p=0.3296 | ρ=0.467 p=0.128 | ρ=0.013 p=0.9261 |

ab – antibody, AGTR2 – Angiotensin-II-Receptor-1 ADRB1 – Beta-1 Adrenergic Receptor, ADRB2 – Beta-2 Adrenergic Receptor, EDNRA – Endothelin Receptor, CHRM3 – Muscarinergic Choline Receptor 3, CHRM4 – Muscarinergic Choline Receptor 4.

In order to elucidate the relationship between the GPCR autoantibodies in Post-COVID and the vascular function, we compared any difference in vasodilatory capacity between groups. Given the numerical trend toward lower FMD values in Post-COVID (Table 2), we subsequently performed a Spearman correlation analysis between the autoantibody concentrations and FMD. EDNRA concentrations were positively associated with FMD (p = 0.0382) in patients with Post-COVID. However, this association did not remain statistically significant after adjustment for covariates including sex, BMI, and vaccination status (p = 0.0745). Furthermore, there was a negative correlation between AGTR2-, ADRB1-, and ADRB1-autoantibodies with central systolic blood pressure (p = 0.0276, p = 0.0027, p = 0.0013, respectively) and a significant negative association between central diastolic blood pressure and anti-ADRB1 and -CHMR4-concentrations (ADRB1 p = 0.0045, CHRM4 p = 0.0086) in patients suffering from Post-COVID. Fig 1 illustrates the positive association of EDNRA and FMD in Post-COVID, Fig 2 the significant associations of GPCR autoantibodies with systolic and diastolic central aortic blood pressure in Post-COVID.

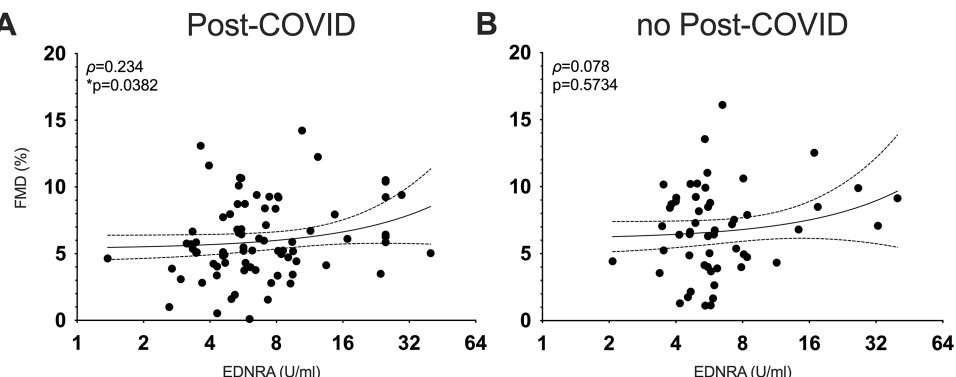

**Fig 1. Correlation analysis of flow-mediated dilation (FMD) with GPCR autoantibodies in Post-COVID and in the control group.** Concentrations of **(A)** EDNRA autoantibodies in the Post-COVID population are positively associated with FMD in Spearman correlation, whereas there is no correlation in the no Post-COVID population **(B)**. Asterisks indicate p-value (**** < 0.0001; *** < 0.001, ** < 0.01, * < 0.05).

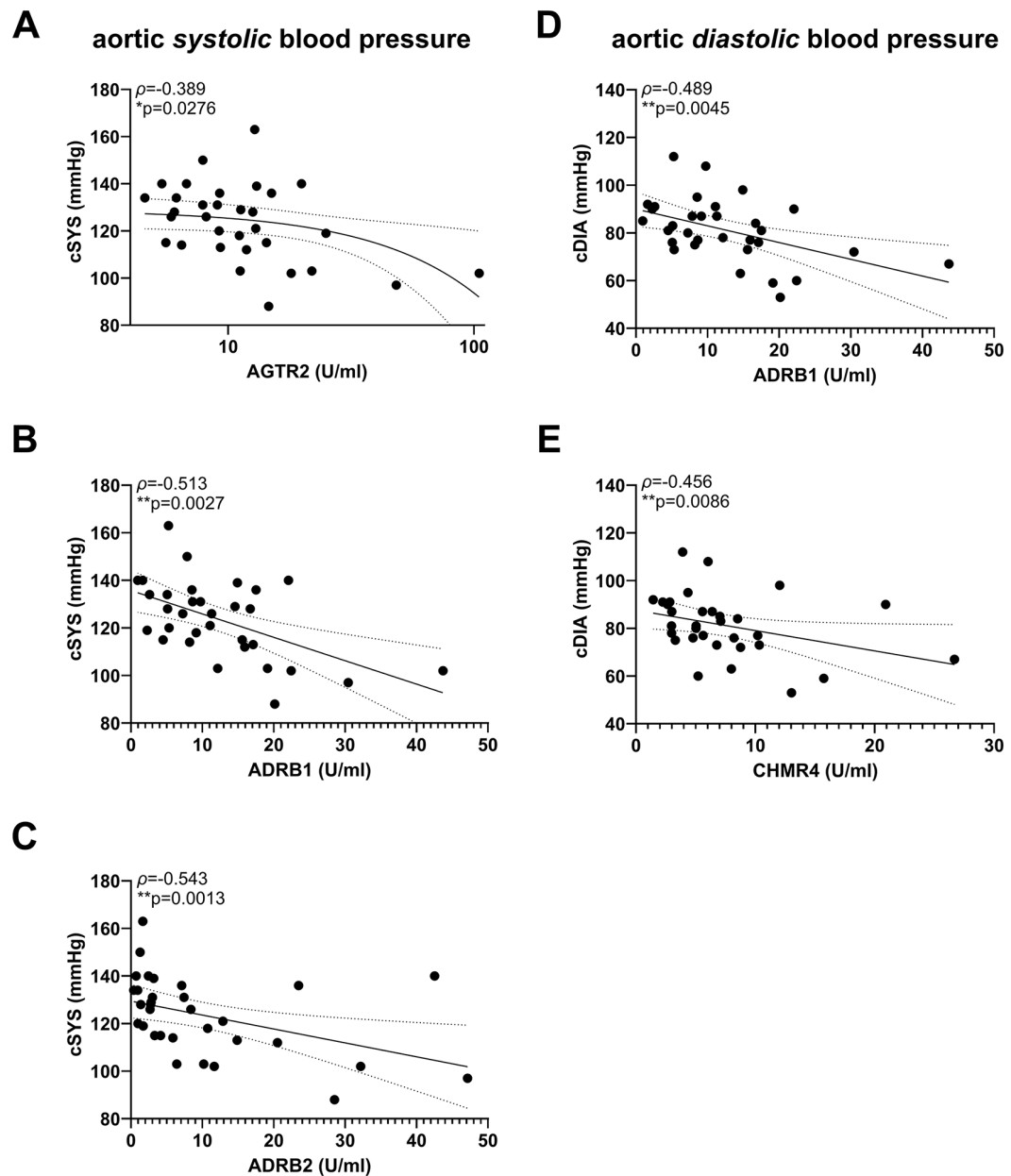

**Fig 2. Correlation analysis of aortic systolic and diastolic blood pressure with GPCR autoantibodies in Post-COVID Concentrations of (A) ATGR2, (B) ADRB1 and (C) ADRB2 autoantibodies are inversely associated with central systolic blood pressure.** Concentrations of **(D)** ADRB1 and **(E)** CHMR4 autoantibodies were inversely associated with central diastolic blood pressure. Asterisk indicate p-value (**** < 0.0001; *** < 0.001, ** < 0.01, * < 0.05). cSYS – central systolic blood pressure; cDIA – central diastolic blood pressure.

In both groups, none of the GPCR autoantibodies were associated with Lp-PLA2 (p > 0.05). We investigated morphological and functional changes in microcirculation as seen by nailfold capillaroscopy. None of the investigated sub-entities (capillary density, vessel diameter variability, elongation, ramification) had a significant association with GPCR autoantibodies in Post-COVID (p > 0.05 each).

## Discussion

"Take Post-COVID condition seriously by urgently investing in research, recovery, and rehabilitation". The present study aims to follow this urgent call from the World Health Organization, who believes millions will be affected in the years to come [18]. The study's findings confirm an increased generation of GPCR autoantibodies in Post-COVID compared to subjects without persisting complaints after SARS-CoV-2 infections. Beyond this descriptive finding, it remains unclear why higher autoantibody levels in Post-COVID patients do not translate into overt differences in vascular parameters when compared to Non–Post-COVID individuals.

Preliminary findings suggest that immune and vascular dysregulations may be involved in the pathophysiology of Post-COVID [19]. We and others have shown that Post-COVID is associated with increased prevalence of GPCR autoantibodies. We have recently demonstrated that the concentrations of these autoantibodies correlate with the intensity of cognitive and physical impairment in Post-COVID [6]. A correlation between levels of autoantibodies against immuno- and vasoregulatory GPCR and cardiovascular clinical severity was found in another study of acute COVID-19 [20]. By induction or alteration of signaling, these autoantibodies can regulate the autonomous nervous system, and the function of immune and endothelial cells function – factors known to be involved in COVID-19 pathogenesis [19]. The present study further contributes to our understanding of these autoantibodies: It shows that GPCR autoantibody concentrations might have an association with markers of vasodilation. FMD is a measure of endothelial function, since the dilation of the brachial artery is mediated by NO release of intact endothelial cells after a transient period of ischemia. In line with this finding, autoantibody concentrations inversely correlate with central systolic blood pressure. There is a notably higher level of ADRB2 autoantibodies in the non-Post-COVID group with at least one positive GPCR autoantibody compared to Post-COVID patients, which may point toward a modulatory or compensatory role of these autoantibodies in maintaining vascular and immune homeostasis. Endogenous autoantibodies against GPCRs (including β2-adrenergic receptors) are known to act as functional modulators of receptor signaling and immune regulation rather than just disease markers [21]. There is a growing body of data on the complex nature of these autoantibodies, which are known to exert agonistic and antagonistic effects on the GPCR, consequently exerting both vasoconstrictor and vasodilatory effects [22].

Endothelin-1 receptor type A (EDNRA) is a G-protein-coupled receptor expressed on the surface of a great variety of cells, e.g., immune cells, vascular smooth muscle cells and endothelial cells [23–25]. Certain autoantibodies are specific for these receptors and can regulate their function, some yielding vasoconstriction through $ET_A$, but also mediating inflammatory signals [26]. On the other hand, there are vasodilator effects as the result of a release of substances like nitric oxide (NO), prostacyclin and endothelium-derived hyperpolarizing factor (EDHF) through $ET_B$-receptor-stimulation [27]. An improved vascular function, measured by predominant NO-mediated dilation, has been described by an increasing concentration of EDNRA autoantibodies in our Post-COVID cohort and might mirror the mechanism described above. Flow mediated dilation slows down the pulse pressure wave, leading to a reduced central blood pressure, as observed in various studies as well as in this very one [28,29]. All the other GPCR autoantibodies, that have been investigated in this study, are involved in vasoregulation as well via either angiotensin receptors, betaadrenergic or muscarinergic receptors [30,31]. Thus, they are of crucial relevance for local vasomotor tone. It has been hypothesized, that alterations in perfusion might be the pathophysiological basis for Post-COVID symptoms like fatigue or mnestic deficits. At first sight, it might appear surprising, that the present study presents an association of GPCR autoantibody concentrations and vasodilation instead of vasoconstriction. Vasodilation, however, can go along with both local hyperemia and systemic hypotension and can thereby alter physiological conditions as well. Thus, the hypothesis of cerebral vasodilation as the pathophysiological basis of migraine was supported for decades. Noteworthy, however, the present study – due to its descriptive character – is not able to prove a causal relation between GPCR autoantibodies, changes in vasoregulation and Post-COVID symptoms. Ultimately, our data raise the question of how increased autoantibody levels in Post-COVID patients can be reconciled with the absence of measurable functional vascular (FMD) differences between groups. One possible

explanation is that autoantibodies may exert harmful or dysregulating effects on the vasculature, but that compensatory mechanisms, such as counterregulatory endothelial or neurohumoral pathways, may mask these effects at the macroscopic level, resulting in preserved FMD and central blood pressure values.

Central aortic blood pressure and FMD are markers of arterial function. The present study investigated capillary function as well by means of nailfold microscopy in the Post-COVID patients. Our findings did not reveal an association between GPCR autoantibody concentrations and capillary function. This also aligns with the hypothysesis, which suggests that autoantibodies might contribute to maintaining vascular homeostasis rather than impairing it. If autoantibodies serve a compensatory or modulatory purpose in Post-COVID patients, they could help preserve vascular parameters within normal ranges, potentially at the expense of other organ systems such as the central nervous system. In this scenario, it would be conceivable that autoantibodies support vascular stability but simultaneously participate in cognitive dysfunction, a hypothesis that warrants explicit consideration in future mechanistic studies.

Microinflammatory processes have been proposed as an alternative pathomechanism in Post-COVID. A recent review investigated inflammatory and vascular biomarkers in Post-COVID. C-reactive protein was found to be increased in COVID-19 survivors, that suffered from Post-COVID [32]. In the present study, we focused on vascular inflammation. Lp-PLA2 concentrations were not associated with GPCR autoantibodies. Our results about nailfold capillaroscopy supports this finding.

A limitation of this study is the limited availability of hemodynamic data in the control group. In addition, the absence of subgroup analyses for potential confounding variables (e.g., COVID-19 severity, vaccination status) also represents a limitation. Therefore, we cannot exclude the possibility that other unmeasured or uncontrolled factors may account for the observed associations. It cannot be ruled out that autoantibodies do not play a functional role in vascular regulation, given that no vascular alterations were observed between Post-COVID and Non–Post-COVID groups. A more rigorous evaluation of vascular parameters alongside autoantibody measurements or a larger sample size would be required to clarify this point.

The present study confirms that Post-COVID is associated with an increased generation of GPCR autoantibodies. It has previously been shown that the concentration of these autoantibodies correlates with neurological symptoms of Post-COVID. Although certain autoantibodies were associated with parameters of vasoregulation, the present data do not provide evidence for a causal, pathogenic, or etiologically relevant role of these autoantibodies in Post-COVID. Future longitudinal and mechanistic studies are required to determine whether GPCR autoantibodies are merely markers of immune dysregulation or actively involved in Post-COVID pathophysiology.

## Author contributions

**Conceptualization:** Felix S. Seibert.

**Data curation:** Felix S. Seibert, Melisa Kurucay, Lea Wiemers.

**Formal analysis:** Felix S. Seibert.

**Investigation:** Melisa Kurucay, Lea Wiemers.

**Methodology:** Ulrik Stervbo, Oliver Sander.

**Resources:** Melisa Kurucay, Lea Wiemers, Ulrik Stervbo, Oliver Sander, Monika Segelbacher, Maximilian Seidel, Sebastian Bertram, Nina Babel, Timm H. Westhoff.

**Supervision:** Felix S. Seibert, Melisa Kurucay, Lea Wiemers, Timm H. Westhoff.

**Writing – original draft:** Felix S. Seibert.

**Writing – review & editing:** Timm H. Westhoff.

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
