## [Decision Letter · Decision Letter 0]

2 May 2025

Dear Dr. Seibert,

Thank you for submitting your manuscript to PLOS ONE. After careful consideration, we feel that it has merit but does not fully meet PLOS ONE’s publication criteria as it currently stands. Therefore, we invite you to submit a revised version of the manuscript that addresses the points raised during the review process.

We look forward to receiving your revised manuscript.

Kind regards,

Eliseo A Eugenin, Ph.D.

Academic Editor

PLOS ONE

Journal Requirements:

2. In this instance it seems there may be acceptable restrictions in place that prevent the public sharing of your minimal data. However, in line with our goal of ensuring long-term data availability to all interested researchers, PLOS’ Data Policy states that authors cannot be the sole named individuals responsible for ensuring data access (http://journals.plos.org/plosone/s/data-availability#loc-acceptable-data-sharing-methods).

Additional Editor Comments:

Dear Dr. Seibert:

Thank you for submitting your manuscript to PLOSone. The manuscript was reviewed by 2 experts in the area with significant concerns and suggestion. The manuscript was label as major revision, and upon corrections will be consider again for publication. Please answer all the comments and questions

Best Regards

Eliseo Eugenin

Reviewers' comments:

Reviewer's Responses to Questions

**Comments to the Author**

1. Is the manuscript technically sound, and do the data support the conclusions?

Reviewer #1: Yes

Reviewer #2: Partly

2. Has the statistical analysis been performed appropriately and rigorously?

Reviewer #1: Yes

Reviewer #2: Yes

3. Have the authors made all data underlying the findings in their manuscript fully available?

Reviewer #1: No

Reviewer #2: Yes

4. Is the manuscript presented in an intelligible fashion and written in standard English?

Reviewer #1: Yes

Reviewer #2: Yes

Reviewer #1: The MS by Seibert et al investigates the role of autoantibodies in long COVID by correlating the content of a set of auto-antibody with various blood parameters in Long COVID vs COVID recovered patients. Authors report a strong negative correlation between CPCR autoantibodies and blood pressure parameters.

This study is well performed and data presented supports author’s conclusions.

Major issues:

1- This study is a follow up of the same group showing a correlation between vasoregulatory GPCR autoantibodies and neuronal disordered. Authors should compare autoantibodies in these two studies, the published one and the present one and see if autoantibodies to GPCR correlates similarly with neuronal symptoms and blood pressure variations.

2- Patients need to be better described: interval between acute COVID and sampling; symptoms developed by Long COVID patients and their absence in Controls, comorbidities and as well as history of autoimmunity in the two groups, etc… and give frequencies in the cohort. Then they could ask if autoantibody (or one type of autoantobody) concentration correlates with some specific symptoms?

3- Although correlation shown in Figure 1 are statistically valid, the intensity of the correlation is not high. One question then the relevance of the correlation. Does this figure include only Long COVID patients? Or also Covid resolved one?

4-Figure 1 and 2: Authors should present the same correlation of COVID recovered patients, especially as the 22% of Covid recovered with autoantibodies have concentration similar to COVID recovered ones, apart for ADRB2 which concentration is much higher in COVID recovered vs long COVID. Authors should comment also this latest difference.

Reviewer #2: A) Regarding the abstract:

In the results section, the authors state that “There was no significant association of any GPCR autoantibody concentration with FMD or Lp-PLA2 in the control group,” but later in the manuscript they mention that no association was found in either group, not only in the controls. This discrepancy should be clarified.

B) Regarding the results:

1. The authors report a positive association between EDNRA autoantibodies and FMD. However, is FMD significantly different between the Post-COVID and the non-Post-COVID groups? A comparison of FMD values between these two groups should be presented first. Only then should the association between EDNRA and FMD in the Post-COVID group be introduced. If no difference exists between the Post-COVID and control group FMD,

a) Is the observed association between FMD and autoantibodies clinically relevant?

b) Could this association be merely reflecting a third, unaccounted-for confounding variable (If no significant differences in FMD are observed between the groups)?

2. Similarly to 1., the authors report a negative association between AGTR2, ADRB1, and ADRB2 autoantibody levels and central systolic blood pressure, as well as a negative association between ADRB1 and CHMR4 autoantibodies and central diastolic blood pressure. However, do baseline vascular parameters differ between the Post-COVID and non-Post-COVID groups? If no data from the control group are available, how do the vascular parameters of the Post-COVID group compare to “normal/expected” values reported in the literature?

3. The authors state that no association was found between Lp-PLA2 levels and Post-COVID symptoms. In the Materials and Methods section, it is mentioned that Lp-PLA2 has been correlated with COVID-19 severity. Was any subgroup analysis performed based on the severity of the acute COVID-19 infection in the Post-COVID group? For example, was an association explored between Lp-PLA2 levels and individuals who had mild COVID-19, and separately for those with moderate COVID-19? The lack of subgroup analysis for potential confounding variables (e.g., COVID-19 severity, vaccination status) could represent a limitation of the study. Such an analysis should be conducted, or otherwise acknowledged as a limitation of the study.

4. The authors mention that no association was found between GPCR autoantibodies and functional or morphological microcirculatory changes. It would be valuable to include figures or tables that represent these negative findings.

C) Regarding the discussion:

The limitations section should consider the possibility that other variables may account for the observed associations."

D) Regarding the tables:

1.- In Table 3, the table legend seems inconsistent with the table content. It states “numbers in brackets indicate…”, but the table does not contain any brackets.

E) Regarding grammar and typography:

There are minor typographical and grammatical errors throughout the manuscript. For example: in the abstract, the word “BACKGROUD” should be corrected to “BACKGROUND”. In Table 1, capitalization should be standardized for terms such as sex, chronic obstructive pulmonary disease, hypercholesterinemia, and smoking.The word “Comborbidities” should be corrected to “Comorbidities”.

**Do you want your identity to be public for this peer review?** For information about this choice, including consent withdrawal, please see our Privacy Policy

Reviewer #1: No

Reviewer #2: **Yes:** Juan Caros Prieto Villalobos

---

## [Author Response · Author response to Decision Letter 1]

29 Sep 2025

A Response To the Reviewers has been uploaded in a separate file.

---

## [Decision Letter · Decision Letter 1]

11 Nov 2025

Dear Dr. Seibert,

Thank you for submitting your manuscript to PLOS ONE. After careful consideration, we feel that it has merit but does not fully meet PLOS ONE’s publication criteria as it currently stands. Therefore, we invite you to submit a revised version of the manuscript that addresses the points raised during the review process.

We look forward to receiving your revised manuscript.

Kind regards,

Eliseo A Eugenin, Ph.D.

Academic Editor

PLOS ONE

Journal Requirements:

Additional Editor Comments:

Dear Dr. Seibert

Thank you for resubmitting your manuscript with the corrections. Despite your effort, several issues still remain. If you are not able to answer the comments or suggestions, please provide an explanation. We expect you to complete all the suggestions and comments. I think all of them are constructive comments.

Best regards

Eliseo Eugenin

Reviewers' comments:

Reviewer's Responses to Questions

**Comments to the Author**

Reviewer #2: All comments have been addressed

Reviewer #3: (No Response)

2. Is the manuscript technically sound, and do the data support the conclusions?

Reviewer #2: Yes

Reviewer #3: Partly

3. Has the statistical analysis been performed appropriately and rigorously?

Reviewer #2: Yes

Reviewer #3: Yes

4. Have the authors made all data underlying the findings in their manuscript fully available?

Reviewer #2: Yes

Reviewer #3: No

5. Is the manuscript presented in an intelligible fashion and written in standard English?

Reviewer #2: Yes

Reviewer #3: Yes

Reviewer #2: General Summary:

Seibert et al. in “Association of G-Protein-Coupled Receptors Autoantibodies with Vasoregulation in Post-COVID” present a cross-sectional study showing that patients with Post-COVID have a higher incidence of autoantibodies compared to non–Post-COVID patients. Furthermore, they report significant associations between these autoantibodies and vascular parameters such as flow-mediated dilation (FMD), aortic systolic blood pressure, and aortic diastolic blood pressure in the Post-COVID group. Finally, they show that no statistically significant association was found between GPCR autoantibody concentrations and FMD or Lp-PLA2 in the control group.

A) General Comments: The authors adequately address all comments raised by Reviewer 2.

Based on the new information provided, the following can be concluded:

1) There is a difference in autoantibody levels between the Post-COVID and Non–Post-COVID groups.

2) There appears to be no significant difference in vascular function between Post-COVID and Non–Post-COVID groups: No statistically significant difference in FMD was found between groups (although a slight trend was observed); No difference in central systolic or diastolic blood pressure was observed (although the statistical power is limited).

3) There is an association between autoantibodies and vascular parameters within the Post-COVID group.

Based on these findings, several hypotheses can be proposed (some options):

Option 1: Autoantibodies may exert a harmful effect on the vasculature, but some unknown compensatory mechanism (not investigated in this study) might be counteracting this effect, resulting in no observable difference in vascular parameters between Post-COVID and Non–Post-COVID groups.

Option 2: Autoantibodies may play a compensatory or regulatory role, meaning that vascular parameters remain within normal ranges because autoantibodies contribute to maintaining homeostasis (e.g., without these autoantibodies, patients might present with hypertension). This could also explain why no differences were found in microcirculatory parameters or Lp-PLA2.

Option 3: Autoantibodies might not play any role in vascular regulation. The fact that vascular parameters are similar between groups despite higher antibody levels in Post-COVID patients suggests that the observed associations could be explained by unmeasured confounding variables.

B) Specific Comments:

Although this is a descriptive study, these possibilities are not clearly articulated in the discussion section. For example:

1) Regarding Option 1: How can the increase in autoantibodies be explained in the absence of changes in vascular parameters between Post-COVID and Non–Post-COVID groups? The manuscript mentions the association with vasodilatory parameters and suggests a potential link to pathological vasodilation, but the newly presented data only show a slight (non-significant) decrease in FMD. How can this be reconciled? This point is not clearly discussed.

2) Regarding Option 2:

Could it be possible that in Post-COVID patients, autoantibodies play a compensatory vascular role but exert harmful effects at the cognitive level? If this is a plausible scenario, it should be explicitly addressed in the discussion, either as a potential mechanism or as something the authors consider unlikely.

3) Regarding Option 3:

The discussion currently states: “Therefore, we cannot exclude the possibility that other unmeasured or uncontrolled factors may account for the observed associations.”

This may not be sufficient. It could be expanded by stating, for example:

“It cannot be ruled out that autoantibodies do not play a functional role in vascular regulation, given that no vascular alterations were observed between Post-COVID and Non–Post-COVID groups. A more rigorous evaluation of vascular parameters alongside autoantibody measurements or a larger sample size would be required to clarify this point.”

Reviewer #3: General comment:

The authors present a cross-sectional study on the involvement of vascular dysfunction in the post-acute COVID-19 syndrome (PACS) which they address as post-COVID. The study cohort and the control cohort are well defined and the data attesting to the presence of vascular dysfunction in post-COVID (PACS) are convincing. The elaborations on a possible etiological role of GPCR-antibodies in post-COVID (PACS) are much less convincing. It remains unclear how the panel of studied GPCR-antibodies was selected (why these and no others ?) and how specific were the observed correlations of antibody titres with hemodynamic parameters for Post-COVID (shown in Fig. 2) as compared to control (not shown). The results on GPCR-antibodies are equivocally discussed and not compared to comparable studies performed in cardiovascular disease not associated to/ following COVID-19. The authors have not entertained the thought that GPCR-antibodies might be used as disease markers, and they seem unaware of prominently published evidence suggesting GPCRS-antibodies as physiological immune-modulators, possibly upregulated in inflammatory diseases in a protective manner (see e.g. Cabral-Marques et al 2018, Nat Commun. 2018;9(1):5224.). The poor quality of the GPCR-antibody part of the study is particularly unfortunate since one is directed towards the issue of GPCR-antibodies in the title. Thus the erroneous impression is created that one is dealing with a study on GPCR-antibodies, while that issue seems just a side-aspect. In summary, the paper could be largely improved by either bringing the part dealing with GPCR-antibodies up to a professional level (some suggestions given below), or by removing the GPCR-antibodies altogether and focusing the work on hemodynamics in post-COVID (PACS).

Details and suggestions

Abstract: The authors pose the initial question “whether detection and concentration of GPCR autoantibodies are related to vascular function in patients with Post-COVID”.

However, they fail to address that question in their final statement (are GPCR autoantibodies related to vascular dysfunction in Post-COVID, and if so, in what manner?). The current final sentence focusses on the role of vascular dysfunction in post-COVID, which is also interesting, but has not been defined as the main goal of the study .

Terminology: The authors are encouraged to use the term “post-acute COVID-19 syndrome (PACS)” or another of the established names of that syndrome (e.g. to be found in refs 1-3) and not invent yet another one (i.e. post-COVID).

GPCR-autoantibodies: By which criteria have the authors selected the panel of GPCR-antibodies investigated in their study. How were the cut-offs determined? How do these cut-offs compare to those used in other studies of GPCR-antibodies in cardiovascular disease (related and unrelated to COVID-19).

In the landmark-mark study on ME/CFS following COVID-19 (ref 4) total interleukin-8 has been identified as a distinctive parameter. Why was that parameter not included in the present study, given that ref 4 was available at the onset of the study.

Vaccination: The authors state in the results section: “In contrast to patients affected by Post-COVID, 88.9% of subjects in the non-Post-COVID group underwent vaccination (p=0.0001).” That seems an interesting observation. I assume that the authors refer to vaccination against the SARS-CoV-2 virus? The statement should be corroborated by more information: How many study patients affected by Post-COVID were not vaccinated? Were there any differences in the vaccination regimes or the type of vaccine administered, etc ? In case that information is hidden somewhere in the tables and I have missed it, a reference should be included. If SARS-CoV-2-vaccination is considered a decisive issue, SARS-CoV-2-virus serology (spike S1-antibodies and nucleo-capsid protein antibodies) should be determined.

Discussion and conclusion: Whether increases in titres of GPCR-antibodies associated with cardiovascular dysfunction are an epiphenomenon, a pathogenic mechanism or a protective reaction of the immune system is a highly controversial issue (see e.g. Zweck et al, 2023, ESC HEART FAILURE DOI:10.1002/ehf2.14293). In this context, the authors discuss their data very selective and quite equivocally. On the one hand they state: “… the present study – … – is not able to prove a causal relation between GPCR autoantibodies, changes in vasoregulation and Post-COVID symptoms.” On the other hand, they claim: “The potential association with vasodilatory parameters … indicates, that GPCR autoantibody mediated dysregulation of vasomotor tone might be a piece in the puzzle in the etiology of Post-COVID.” In my opinion, the two statements do not entirely conform to the state of knowledge, and, most notably, contradict each other. Moreover, the second one is pure speculation. Either the data show an association of uncertain pathogenic relevance (i.e. an epiphenomenon) or they provide evidence of an etiologically relevant regulatory or pathogenic mechanism. There is no middle way. The authors should make up their mind or be silent.

**Do you want your identity to be public for this peer review?** For information about this choice, including consent withdrawal, please see our Privacy Policy

Reviewer #2: No

Reviewer #3: No

---

## [Author Response · Author response to Decision Letter 2]

29 Dec 2025

RESPONSE TO REVIEWERS

Dear Editor,

Thank you very much for handling our manuscript and for the reviewers’ insightful comments. We have carefully addressed all points raised, as detailed below.

Editor Comments

Reviewer #2: General Summary:

Seibert et al. in “Association of G-Protein-Coupled Receptors Autoantibodies with Vasoregulation in Post-COVID” present a cross-sectional study showing that patients with Post-COVID have a higher incidence of autoantibodies compared to non–Post-COVID patients. Furthermore, they report significant associations between these autoantibodies and vascular parameters such as flow-mediated dilation (FMD), aortic systolic blood pressure, and aortic diastolic blood pressure in the Post-COVID group. Finally, they show that no statistically significant association was found between GPCR autoantibody concentrations and FMD or Lp-PLA2 in the control group.

A) General Comments: The authors adequately address all comments raised by Reviewer 2. Based on the new information provided, the following can be concluded:

1) There is a difference in autoantibody levels between the Post-COVID and Non–Post-COVID groups.

2) There appears to be no significant difference in vascular function between Post-COVID and Non–Post-COVID groups: No statistically significant difference in FMD was found between groups (although a slight trend was observed); No difference in central systolic or diastolic blood pressure was observed (although the statistical power is limited).

3) There is an association between autoantibodies and vascular parameters within the Post-COVID group.

Based on these findings, several hypotheses can be proposed (some options):

Option 1: Autoantibodies may exert a harmful effect on the vasculature, but some unknown compensatory mechanism (not investigated in this study) might be counteracting this effect, resulting in no observable difference in vascular parameters between Post-COVID and Non–Post-COVID groups.

Option 2: Autoantibodies may play a compensatory or regulatory role, meaning that vascular parameters remain within normal ranges because autoantibodies contribute to maintaining homeostasis (e.g., without these autoantibodies, patients might present with hypertension). This could also explain why no differences were found in microcirculatory parameters or Lp-PLA2.

Option 3: Autoantibodies might not play any role in vascular regulation. The fact that vascular parameters are similar between groups despite higher antibody levels in Post-COVID patients suggests that the observed associations could be explained by unmeasured confounding variables.

B) Specific Comments: Although this is a descriptive study, these possibilities are not clearly articulated in the discussion section. For example:

1) Regarding Option 1: How can the increase in autoantibodies be explained in the absence of changes in vascular parameters between Post-COVID and Non–Post-COVID groups? The manuscript mentions the association with vasodilatory parameters and suggests a potential link to pathological vasodilation, but the newly presented data only show a slight (non-significant) decrease in FMD. How can this be reconciled? This point is not clearly discussed.

2) Regarding Option 2: Could it be possible that in Post-COVID patients, autoantibodies play a compensatory vascular role but exert harmful effects at the cognitive level? If this is a plausible scenario, it should be explicitly addressed in the discussion, either as a potential mechanism or as something the authors consider unlikely.

3) Regarding Option 3: The discussion currently states: “Therefore, we cannot exclude the possibility that other unmeasured or uncontrolled factors may account for the observed associations.” This may not be sufficient. It could be expanded by stating, for example: “It cannot be ruled out that autoantibodies do not play a functional role in vascular regulation, given that no vascular alterations were observed between Post-COVID and Non–Post-COVID groups. A more rigorous evaluation of vascular parameters alongside autoantibody measurements or a larger sample size would be required to clarify this point.”

ANSWER: We thank the Reviewer for his thorough and constructive comments. Each of the points raised has been carefully addressed and implemented in the final version of the manuscript.

Reviewer #3: General comment:

The authors present a cross-sectional study on the involvement of vascular dysfunction in the post-acute COVID-19 syndrome (PACS) which they address as post-COVID. The study cohort and the control cohort are well defined and the data attesting to the presence of vascular dysfunction in post-COVID (PACS) are convincing. The elaborations on a possible etiological role of GPCR-antibodies in post-COVID (PACS) are much less convincing. It remains unclear how the panel of studied GPCR-antibodies was selected (why these and no others ?) and how specific were the observed correlations of antibody titres with hemodynamic parameters for Post-COVID (shown in Fig. 2) as compared to control (not shown). The results on GPCR-antibodies are equivocally discussed and not compared to comparable studies performed in cardiovascular disease not associated to/ following COVID-19. The authors have not entertained the thought that GPCR-antibodies might be used as disease markers, and they seem unaware of prominently published evidence suggesting GPCRS-antibodies as physiological immune-modulators, possibly upregulated in inflammatory diseases in a protective manner (see e.g. Cabral-Marques et al 2018, Nat Commun. 2018;9(1):5224.). The poor quality of the GPCR-antibody part of the study is particularly unfortunate since one is directed towards the issue of GPCR-antibodies in the title. Thus the erroneous impression is created that one is dealing with a study on GPCR-antibodies, while that issue seems just a side-aspect. In summary, the paper could be largely improved by either bringing the part dealing with GPCR-antibodies up to a professional level (some suggestions given below), or by removing the GPCR-antibodies altogether and focusing the work on hemodynamics in post-COVID (PACS).

Details and suggestions

Abstract: The authors pose the initial question “whether detection and concentration of GPCR autoantibodies are related to vascular function in patients with Post-COVID”. However, they fail to address that question in their final statement (are GPCR autoantibodies related to vascular dysfunction in Post-COVID, and if so, in what manner?). The current final sentence focusses on the role of vascular dysfunction in post-COVID, which is also interesting, but has not been defined as the main goal of the study

ANSWER: We thank the Reviewer for his constructive comment. We changed the abstract accordingly.

Terminology: The authors are encouraged to use the term “post-acute COVID-19 syndrome (PACS)” or another of the established names of that syndrome (e.g. to be found in refs 1-3) and not invent yet another one (i.e. post-COVID).

ANSWER: We adopt the term “Post-COVID” in accordance with leading health authorities, including the WHO, ECDC, and several others. We totally agree with the REVIEWER, it is crucial not to introduce new terminology for officially recognized health conditions. For reference, please consult the following sources:

WHO: https://www.who.int/news-room/fact-sheets/detail/post-covid-19-condition-(long-covid)

ECDC: https://www.ecdc.europa.eu/en/infectious-disease-topics/z-disease-list/covid-19/factsheet-covid-19

GPCR-autoantibodies: By which criteria have the authors selected the panel of GPCR-antibodies investigated in their study. How were the cut-offs determined? How do these cut-offs compare to those used in other studies of GPCR-antibodies in cardiovascular disease (related and unrelated to COVID-19).

ANSWER: The GPCR autoantibody panel was selected based on prior biological relevance and established use in the literature. We applied a commercially available GPCR autoantibody panel that has been used in our previous work (DOI: 10.1016/j.autrev.2023.103445), cut-offs for the detection have been chosen according to the manufacturer. A correlation between levels of autoantibodies against immuno- and vasoregulatory GPCR and cardiovascular clinical severity was found in another study of acute COVID-19 (Cabral-Marques O et al. Nat Commun 2022;13:1220). By induction or alteration of signaling, these autoantibodies can regulate the autonomous nervous system, and the function of immune and endothelial cells function – factors known to be involved in COVID-19 pathogenesis (Sotzny F et al. Front Immunol 2022;13:981532. We thank the Reviewer and adjusted our discussion in the revised version of the manuscript.

In the landmark-mark study on ME/CFS following COVID-19 (ref 4) total interleukin-8 has been identified as a distinctive parameter. Why was that parameter not included in the present study, given that ref 4 was available at the onset of the study.

ANSWER: IL-8 was not included in the scope of the present study at the time of conceptualization. However, an in-depth immunological characterization with particular emphasis on the dynamics of interleukin levels would represent a highly relevant avenue for future research.

Vaccination: The authors state in the results section: “In contrast to patients affected by Post-COVID, 88.9% of subjects in the non-Post-COVID group underwent vaccination (p=0.0001).” That seems an interesting observation. I assume that the authors refer to vaccination against the SARS-CoV-2 virus? The statement should be corroborated by more information: How many study patients affected by Post-COVID were not vaccinated? Were there any differences in the vaccination regimes or the type of vaccine administered, etc? In case that information is hidden somewhere in the tables and I have missed it, a reference should be included. If SARS-CoV-2-vaccination is considered a decisive issue, SARS-CoV-2-virus serology (spike S1-antibodies and nucleo-capsid protein antibodies) should be determined.

ANSWER: We thank the reviewer for this important and constructive comment. Yes, the term vaccination in our manuscript refers exclusively to vaccination against SARS-CoV-2. We agree that this point requires clarification and additional detail. In our cohort, 40 out of 80 patients with Post-COVID (50%) had received at least one SARS-CoV-2 vaccination prior to study inclusion, whereas 48 out of 54 individuals (88.9%) in the non-Post-COVID group were vaccinated, resulting in the reported significant difference (p = 0.0001; Table 1). We explicitly stated these absolute numbers in the Results section and added a direct reference to Table 1 to improve clarity in the revised version of the manuscript. Information on vaccination regimens (number of doses, timing in relation to infection, or specific vaccine types) was not collected in a standardized manner in this study and therefore could not be analyzed reliably. Consequently, we are unable to assess potential differences in vaccination schemes or vaccine platforms between groups. This represents a limitation of our study, which we is acknowledge in the Discussion section.

Regarding SARS-CoV-2 serology, we agree with the reviewer that the determination of spike S1 and nucleocapsid antibodies would provide valuable additional information, particularly for disentangling effects of vaccination versus natural infection. However, SARS-CoV-2 serological testing was not part of the predefined study protocol and serum samples were exclusively used for GPCR autoantibody analyses. Therefore, serological data are not available for the present cohort.

Discussion and conclusion: Whether increases in titres of GPCR-antibodies associated with cardiovascular dysfunction are an epiphenomenon, a pathogenic mechanism or a protective reaction of the immune system is a highly controversial issue (see e.g. Zweck et al, 2023, ESC HEART FAILURE DOI:10.1002/ehf2.14293). In this context, the authors discuss their data very selective and quite equivocally. On the one hand they state: “… the present study – … – is not able to prove a causal relation between GPCR autoantibodies, changes in vasoregulation and Post-COVID symptoms.” On the other hand, they claim: “The potential association with vasodilatory parameters … indicates, that GPCR autoantibody mediated dysregulation of vasomotor tone might be a piece in the puzzle in the etiology of Post-COVID.” In my opinion, the two statements do not entirely conform to the state of knowledge, and, most notably, contradict each other. Moreover, the second one is pure speculation. Either the data show an association of uncertain pathogenic relevance (i.e. an epiphenomenon) or they provide evidence of an etiologically relevant regulatory or pathogenic mechanism. There is no middle way. The authors should make up their mind or be silent.

ANSWER: The reviewer is correct in pointing out that our current wording conveys an ambiguity that may appear contradictory. We adjusted our statement in the revised version of the manuscript.

---

## [Decision Letter · Decision Letter 2]

4 Feb 2026

Association of G-Protein-Coupled Receptor autoantibodies with vasoregulation in Post-COVID

PONE-D-25-02805R2

Dear Dr. Seibert,

We’re pleased to inform you that your manuscript has been judged scientifically suitable for publication and will be formally accepted for publication once it meets all outstanding technical requirements.

Kind regards,

Eliseo A Eugenin, Ph.D.

Academic Editor

PLOS One

Additional Editor Comments (optional):

Dear Dr. Seibert

Thank you for including all the suggestions and comments. Our aapologize for the extended time for the review process.

Best

Eliseo Eugenin

Reviewers' comments:

Reviewer's Responses to Questions

**Comments to the Author**

Reviewer #2: All comments have been addressed

Reviewer #3: All comments have been addressed

2. Is the manuscript technically sound, and do the data support the conclusions?

Reviewer #2: Yes

Reviewer #3: Yes

3. Has the statistical analysis been performed appropriately and rigorously?

Reviewer #2: Yes

Reviewer #3: Yes

4. Have the authors made all data underlying the findings in their manuscript fully available?

Reviewer #2: Yes

Reviewer #3: Yes

5. Is the manuscript presented in an intelligible fashion and written in standard English?

Reviewer #2: Yes

Reviewer #3: Yes

Reviewer #2: Seibert et al., in “Association of G-Protein-Coupled Receptors Autoantibodies with Vasoregulation in Post-COVID” have satisfactorily addressed all the comments raised in the previous review, clearly outlining the potential implications of their results as well as the associated limitations. I have no additional comments. My congratulations to the authors.

Reviewer #3: The manuscript has been much improved by the revision.

This reviewer would like to express his gratitude to the author for having properly addressed all the suggestions.

**Do you want your identity to be public for this peer review?** For information about this choice, including consent withdrawal, please see our Privacy Policy

Reviewer #2: No

Reviewer #3: No

---

## [Editor Report · Acceptance letter]

PONE-D-25-02805R2

PLOS One

Dear Dr. Seibert,

I'm pleased to inform you that your manuscript has been deemed suitable for publication in PLOS One. Congratulations! Your manuscript is now being handed over to our production team.

Kind regards,

on behalf of

Dr. Eliseo A Eugenin

Academic Editor

PLOS One